# Detrimental impact of sulfide on the seagrass *Zostera marina* in dark hypoxia

Harald Hasler-Sheetal [1,2]*

**1** Nordcee, University of Southern Denmark, Odense M, Denmark, **2** VILLUM Center for Bioanalytical Sciences, University of Southern Denmark, Odense M, Denmark

* hasler@sdu.dk

**Data Availability Statement:** All relevant data are within the paper and its Supporting Information files.

**Funding:** The author received no specific funding for this work.

## Abstract

Sulfide poisoning, hypoxia events, and reduced light availability pose threats to marine ecosystems such as seagrass meadows. These threats are projected to intensify globally, largely due to accelerating eutrophication of estuaries and coastal environments. Despite the urgency, our current comprehension of the metabolic pathways that underlie the deleterious effects of sulfide toxicity and hypoxia on seagrasses remains inadequate. To address this knowledge gap, I conducted metabolomic analyses to investigate the impact of sulfide poisoning under dark-hypoxia in vitro conditions on *Zostera marina*, a vital habitat-forming marine plant. During the initial 45 minutes of dark-hypoxia exposure, I detected an acclimation phase characterized by the activation of anaerobic metabolic pathways and specific biochemical routes that mitigated hypoxia and sulfide toxicity. These pathways served to offset energy imbalances, cytosolic acidosis, and sulfide toxicity. Notably, one such route facilitated the transformation of toxic sulfide into non-toxic organic sulfur compounds, including cysteine and glutathione. However, this sulfide tolerance mechanism exhibited exhaustion post the initial 45-minute acclimation phase. Consequently, after 60 minutes of continuous sulfide exposure, the sulfide toxicity began to inhibit the hypoxia-mitigating pathways, culminating in leaf senescence and tissue degradation. Utilizing metabolomic approaches, I elucidated the intricate metabolic responses of seagrasses to sulfide toxicity under in vitro dark-hypoxic conditions. My findings suggest that future increases in coastal eutrophication will compromise the resilience of seagrass ecosystems to hypoxia, primarily due to the exacerbating influence of sulfide.

## Introduction

Seagrasses are marine angiosperms that flourish in sunlit marine sediments, which frequently experience anoxic conditions and the omnipresence of phytotoxic sulfide [1, 2]. In marine sediments, sulfide is preliminary generated through dissimilatory sulfate reduction, a predominant microbial process that oxidizes organic matter via the reduction of sulfate to sulfide [3]. Both sulfate and organic matter are present in surplus in anthropogenically impacted marine ecosystems, thereby facilitating elevated concentrations of detrimental sulfide in seagrass habitats [1]. Under hypoxic conditions, gaseous sulfide intrudes the plant roots and diffuses via extensive aerenchyma into the leaves [4] eventually causing deleterious effects [5–7]. In eukaryotes,

**Competing interests:** The authors have declared that no competing interests exist.

including seagrasses, sulfide inhibits key enzymes in aerobic metabolisms and reduces anaerobic respiration [5, 8, 9]. However, sulfide rapidly oxidizes to non-toxic compounds in the presence of oxygen. Consequently, under normoxic conditions, seagrasses deter sulfide intrusion through radial oxygen loss (ROL) from the roots [4]: During photoperiods, oxygen generated through photosynthesis diffuses via the aerenchyma to the roots [10], where it is released into the sediment to establish an oxic micro-shield, thereby oxidizing toxic sulfide to non-toxic compounds before reaching the root surfaces [10, 11]. In absence of light, the internal oxygen reservoir within the aerenchyma is rapidly (within a few minutes) depleted due to respiration [12], and the plant's internal oxygen pool is sustained exclusively by oxygen diffusion from the water column [13]. Under dark hypoxic conditions, sulfide might intrude into the roots and diffuse into leaves, potentially causing harm upon the exhaustion of tolerance mechanisms [6, 14]. Besides ROL mediated sulfide oxidation (a sulfide avoidance mechanism), seagrasses also metabolize infiltrating sulfide into non-toxic organic sulfur compounds as a tolerance mechanism [14]. Remarkably, is a ubiquitous phenomenon across all seagrass species [1] indicating the universal presence of sulfide avoidance and tolerance mechanisms [14].

While the metabolic adaptations of seagrasses to hypoxia are well-characterized [15], the specific impact of sulfide toxicity remains largely unexplored. Previous studies on the effect of darkness and oxygen deficit on seagrasses have shown that oxidative respiration ceases, evidenced by a non-functioning TCA cycle, and anaerobic fermentation becomes essential to meet energy demands [16]. To mitigate the negative effects of anaerobic metabolism, seagrasses express metabolic shunts to alleviate acidosis and energy deficits [15].

Martin and Maricle [9] exposed various terrestrial and wetland plants to increased soil sulfide levels and observed sulfide-related inhibition of cytochrome c oxidase, along with reduced efficiency of alcohol dehydrogenase in roots. Interestingly, halophytic marsh plants demonstrated substantial sulfide tolerance mechanisms compared to other terrestrial plants [9] and similar mechanisms might be present in seagrasses. Given that both oxygen deprivation and sulfide poisoning inhibit oxidative respiration in eukaryotes and seagrasses [6, 15, 17], similar deleterious metabolic patterns might occur under hypoxia and sulfide intrusion. The cause of seagrass die-offs, whether it is hypoxia or sulfide, or a combination of both, has been a long-standing debate. Furthermore, the molecular basis of how seagrass metabolism is affected by these factors remains unclear.

Metabolomics offers a comprehensive approach to identify and quantify the entire metabolome, thereby reflecting the real-time physiological state of an organism. As such, metabolomics serves as an invaluable tool for investigating the impact of sulfide toxicity on seagrasses by quantifying their adaptive, tolerative, and responsive capacities to fluctuating sulfide levels. In this study, I hypothesize that sulfide toxicity uniquely perturbs seagrass metabolism compared to hypoxia, thereby instigating a distinct reconfiguration of the metabolic network. To test this, I examined whether phytotoxic sulfide concentrations modulate the metabolic pathways and molecular switches in seagrass leaves. Employing metabolomic techniques, I assessed the perturbations in primary energy metabolism, nitrogen metabolism, sulfur metabolism, and tissue degradation, aiming to elucidate the predominant metabolic consequences of sulfide toxicity in seagrasses.

## Material and methods

### Study system

The seagrass species chosen for this study is *Zostera marina L.*, commonly found in estuaries and shallow coastal waters of the temperate Northern Hemisphere [18]. Plant and water samples were collected from the Danish straits, a region connecting the Baltic Sea to the North Sea.

This region is characterized by *Z. marina* habitats with high sulfide levels due to eutrophication and turbidity in Danish coastal waters [1, 19, 20].

## Specimen collection and exposure

To investigate the metabolic ramifications of sulfide intrusion on *Z. marina*, I designed a hydroponic indoor mesocosm experiment under conditions of darkness and hypoxia (oxygen < 2% of air saturation), while manipulating rhizosphere sulfide concentrations (high vs. none) (Fig 1). Seawater and apical seagrass shoots were collected from a shallow coastal bay at Svenstrup Strand, Denmark (55°28′07.1″N 9°45′16.4″E). Following an 8-day acclimation period to laboratory conditions, the samples were transplanted into controlled, recirculating rhizosphere growth chambers situated within 60L seawater aquaria, as elaborated in Pedersen and Kristensen [7]. This experimental setup allowed me to independently manipulate the sulfide concentrations in the rhizosphere, distinct from those in the phyllosphere (water column). The anoxic rhizosphere chambers were maintained at a sulfide concentration of $2.14 \pm 0.03$ mM $H_2S$ by adding $Na_2S$, while the control chambers remained anoxic and sulfide-free (N = 6 and analyzed according to Cline [21]. To initiate the experiment, the oxygen levels in the water column (phyllosphere) were set to $1.8 \pm 1.4\%$ of air saturation by continuous $N_2$ purging.

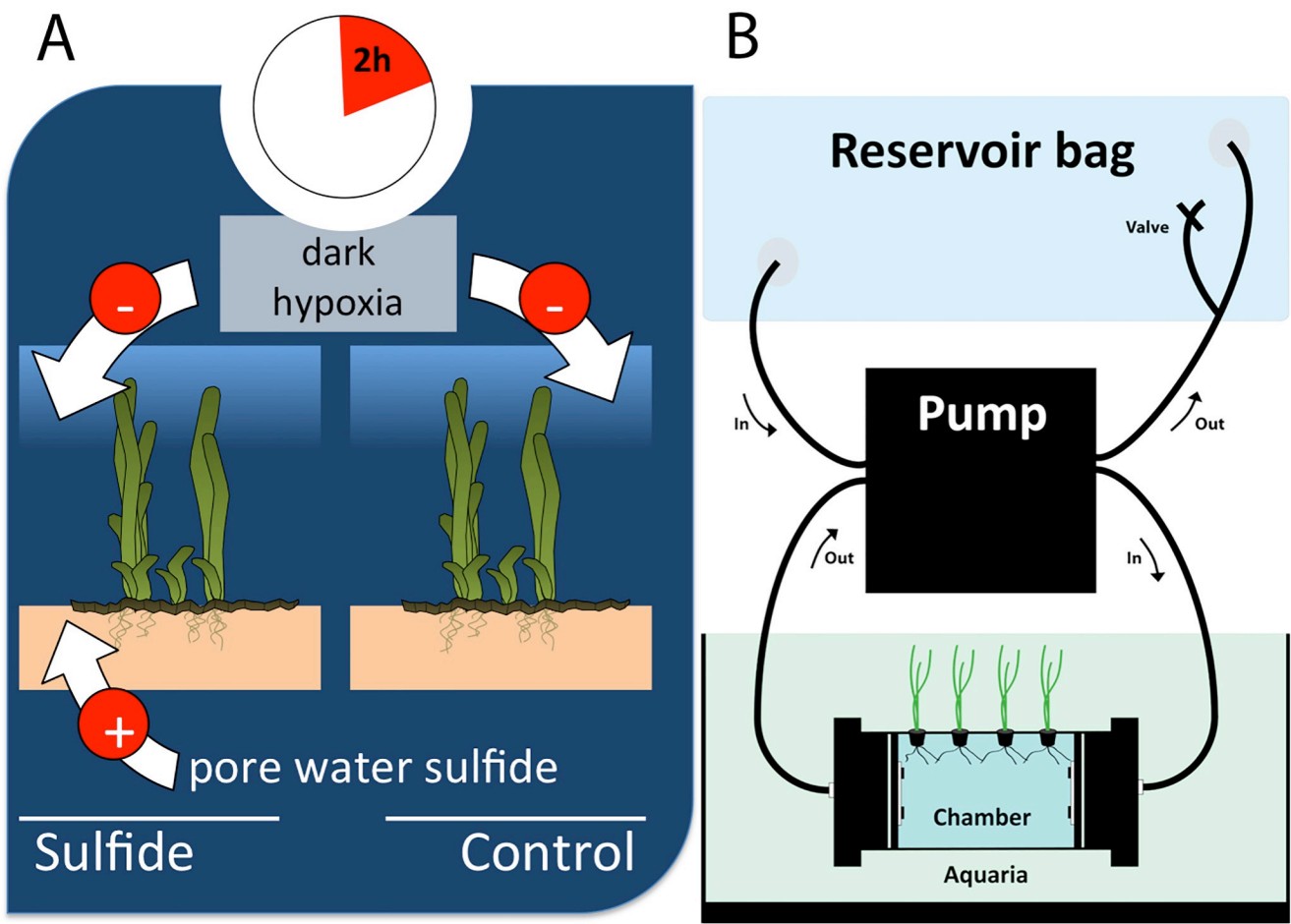

**Fig 1. Experimental design.**

Photosynthetic oxygen production was inhibited by covering the aquaria with black foil Oxygen levels were continuously monitored with a YSI ProODO optical DO sensor (YSI Inc., Yellow Springs, Ohio, USA). Samples were randomly taken (N = 6) at 15 min intervals over a 2-hour period from the start of the experiment yielding a total of 10 time points. Macroscopic epiphytes were manually removed, and the plants were separated into leaves, rhizome, and roots. To preserve the metabolites, the samples were instantaneously flash-frozen in liquid nitrogen (<15 seconds). Subsequently, they were then lyophilized for 48 hours, homogenized in a ball mill for 4 minutes at 20 Hz, and prepared for further processing.

*Zostera marina* shoots were kept under dark hypoxic conditions (A) in recirculating rhizosphere growth chambers placed in 60L seawater aquaria (B). *Z. marina* leaves were exposed to hypoxic oxygen concentrations of 1.8 ± 1.4% of air saturation in the seawater aquaria. I kept the rhizosphere anoxic and sulfidic (2.14 ± 0.01 mM $H_2S$) or sulfide free respectively. I harvested six control and six treatment shoots at 15-minute intervals over a 2-hour duration, yielding a total of ten time points and 120 samples.

## Metabolite profiling

All solvents and tubes were pre-cooled to -20˚C. Ten mg of lyophilized and homogenized plant material were extracted for 15 min on ice in 1 ml methanol/acetonitrile/water (2:2:1 [v/v/v]) spiked with 1μg of $^{13}C_6$-Sorbitol and 1μg Reserpine per sample as internal standards, The extract was subsequently centrifuged at 14,000×g for 4 minutes. The supernatants (900 μl) were transferred to a test tube and partitioned into two 350 μl aliquots for LC-MS analysis and two 100 μl aliquots for GC-MS analysis. The samples were then dried in a speed-vac overnight. For LC-MS analysis, the dried samples were resuspended in 100 μl of LC-solvent A (0.1% formic acid in water). GC-MS analysis was conducted following derivatization using a GC-7200-QTOF-MS system (Agilent Technologies, Santa Clara, CA, USA) following the method described by Hasler-Sheetal, Castorani [22] with slight modifications. Metabolomics analysis was performed using a 1290 quaternary UHPLC system (Agilent Technologies, Santa Clara, CA, USA) and an Agilent 6530 quadropole-time of flight (Q-TOF) mass spectrometer (MS) with an ESI source. Metabolite separation was achieved by injecting 3 μl of the sample onto a reversed-phase column (Agilent Zorbax EclipsePlus C18; 100x2.1 mm, 1.8 μm) maintained at 40˚C. The mobile phase consisted of solvent A (0.1% formic acid in water) and solvent B (0.1% formic acid in acetonitrile). The gradient elution program, flow rate, other settings for LC-QTOF-MS and GC-QTOF-MS analysis and data processing of LC-QTOF-MS and GC-QOF-MS, and data analysis were conducted as described in Hasler-Sheetal, Castorani [22]. In brief, the effects of sulfide concentration over time were compared using analysis of variance (ANOVA; α = 0.05) and Tukey's posthoc test. I applied a false discovery rate correction using the Benjamini–Hochberg method, an adjusted p value of <0.05 was considered significant. Detailed results, including ANOVA outcomes and metabolite levels, are presented in the supplementary materials.

## Results and discussion

I investigated the in vitro metabolic response of the seagrass *Z. marina* to sulfide intrusion under conditions of darkness and hypoxia. The seagrass metabolome was monitored at ten distinct time points over a 2-hour period. Prior research has established that for sulfide to intrude seagrass roots, the oxygen concentration in the overlying water must fall below 35% of air saturation [4, 7, 13]. In our experimental setup, the oxygen levels in the water column were markedly low, at 1.8 ± 1.4% of air saturation. This would have led to rapid depletion of oxygen in the aerenchyma [12]., thereby precluding any "external" oxidation of sulfide to non-toxic

sulfur compounds in both the rhizosphere and aerenchyma. Consequently, this facilitated substantial sulfide intrusion into the seagrass tissues.

Sulfide has been suggested to function as a signaling molecule that modulates physiological processes advantageous for plant performance, particularly at low micromolar concentrations ranging between 10–100 μM [23, 24]. This regulatory role of sulfide in metabolism could potentially extend to seagrasses, which persist despite enduring continuous sulfide intrusion into their tissues at concentrations as high as 325 μM in natural stands [1, 14]. However, the sulfide levels in our study were one order of magnitude higher, specifically $2.14 \pm 0.03$ mM $H_2S$, thereby negating any putative beneficial physiological effects. Sulfide levels exceeding 2 mM have been reported to be lethal for seagrasses and eukaryotes in general [5, 7].

## Metabolic fingerprinting in response to sulfide

I conducted UPLC-MS analysis after extraction, detecting a total of 64,283 mass spectral features. After data processing, I identified 4,303 reproducible metabolite entities from *Zostera marina* leaves. These metabolites, which were present in 80% of the quality control samples and showed a coefficient of variation (CV) < 35%, offer a comprehensive representation of the *Z. marina* metabolome [22]. Without further annotation, these metabolites were used for metabolomic fingerprinting. Using multivariate analysis, specifically principal component analysis (PCA), I analyzed the large dataset comprising 120 samples and 4,303 metabolite entities. PCA was employed to reduce dimensionality and visualize the temporal effect of sulfide exposure on the *Z. marina* metabolome. As depicted in Fig 2, the results reveal a distinct clustering of samples based on treatment and time-specific metabolic responses.

The primary variation observed in the PCA analysis was the temporal progression of anoxia, as evidenced by the time-dependent modulation of the *Z. marina* metabolome. This is illustrated by the separation of sample groups along principal component 1, which accounts for 28.7% of the total variation in the dataset. These observations are congruent with our previous studies highlighting the substantial influence of hypoxia on *Z. marina* metabolism [15, 22].

During the experiment's initial phase (15, 30, and 45 minutes), samples cluster together along component 2, explaining 18.2% of the dataset's variation. This indicates an initial metabolic response of seagrass leaves to dark hypoxia that differs from the response observed in the late phase (60–120 minutes) (Fig 2). In this late phase, a separation between control and sulfide-exposed samples is observed along component 3, accounting for 12.12% of the dataset's variation. This temporal offset between sulfide-exposed and control samples implies that sulfide exposure enhances metabolomic reprogramming and may reflect an early onset of degradation processes related to sulfide exposure.

Distinct sets of metabolites are responsible for the observed separation on each component (Fig 2). This highlights the unique physiological responses of *Z. marina* during the initial and late phases. Specifically, metabolites associated with energy and amino acid metabolism govern the separation during the initial phase. In contrast, metabolites liked to cellular degradation contribute to the separation observed during the late phase.

These findings corroborate earlier research that highlighted an initial response phase to hypoxia [15] and underscore the role of hypoxia as a prerequisite for sulfide-induced tissue degradation [22, 25]. To gain a deeper understanding of the potential effects of hypoxia and sulfide exposure, I further investigated primary energy metabolism. This included a detailed analysis of the associated amino acid metabolism, sulfur metabolism, and the processes leading to leaf senescence and degradation.

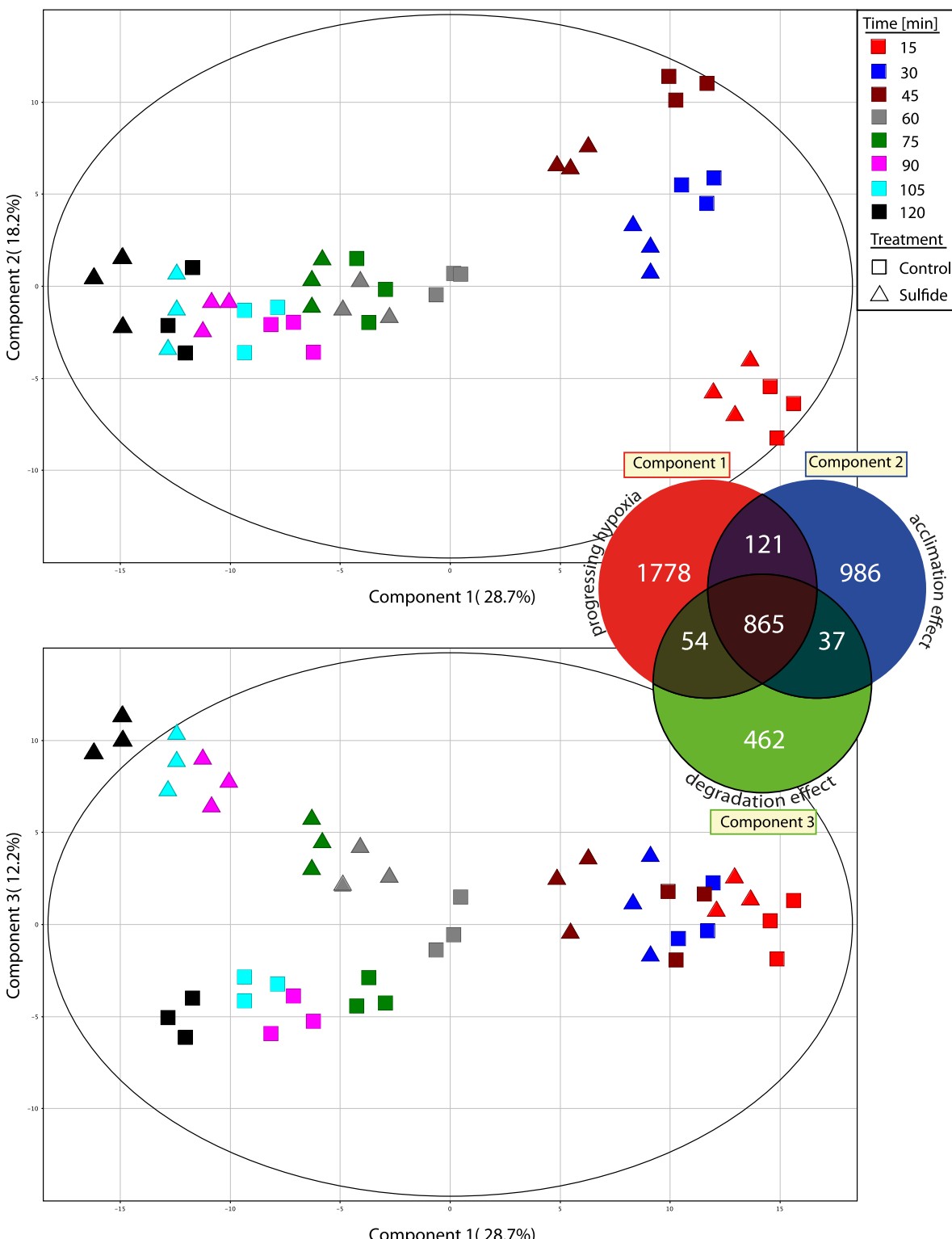

**Fig 2. Metabolomic fingerprinting of Zostera marina leaves over time using PCA.** PCA scores plot that captures the time-related metabolomic fingerprinting of compounds found in Zostera marina leaves when exposed to varying sulfide levels over a span of 120 minutes. The upper panel shows component 1 vs component 2 and the lower panel shows component 2 vs component 3. Triangles indicate samples under high sulfide levels (> 2 mM H₂S) and squares samples under sulfide absence; different colors indicate varying time points. The Venn diagram indicates the number of metabolites with high influence on sample separation on the respective component.

## Energy metabolism and mitigation

The exposure to hypoxia and sulfide distinctly influenced the metabolites associated with primary energy metabolism. This encompasses pathways such as the TCA cycle, glycolysis, and associated amino acid pathways. Notably, these effects varied between the initial (first 45 minutes) and latter stages of the experiment (60 to 120 minutes) (Fig 3). During the initial phase there was a pronounced response of all observed metabolites. However metabolic response in the latter stages was more subdued. These observations align with the pattern seen in the PCA plot (Fig 2) and corroborate prior research that highlighted an initial acclimation response phase to dark hypoxia [15].

During conditions of dark hypoxia, a general response was observed, characterized by a decline in carbohydrate concentration. This is likely attributed to the cessation of photosynthesis combined with an increased glycolysis (Pasteur effect). This combination results in a

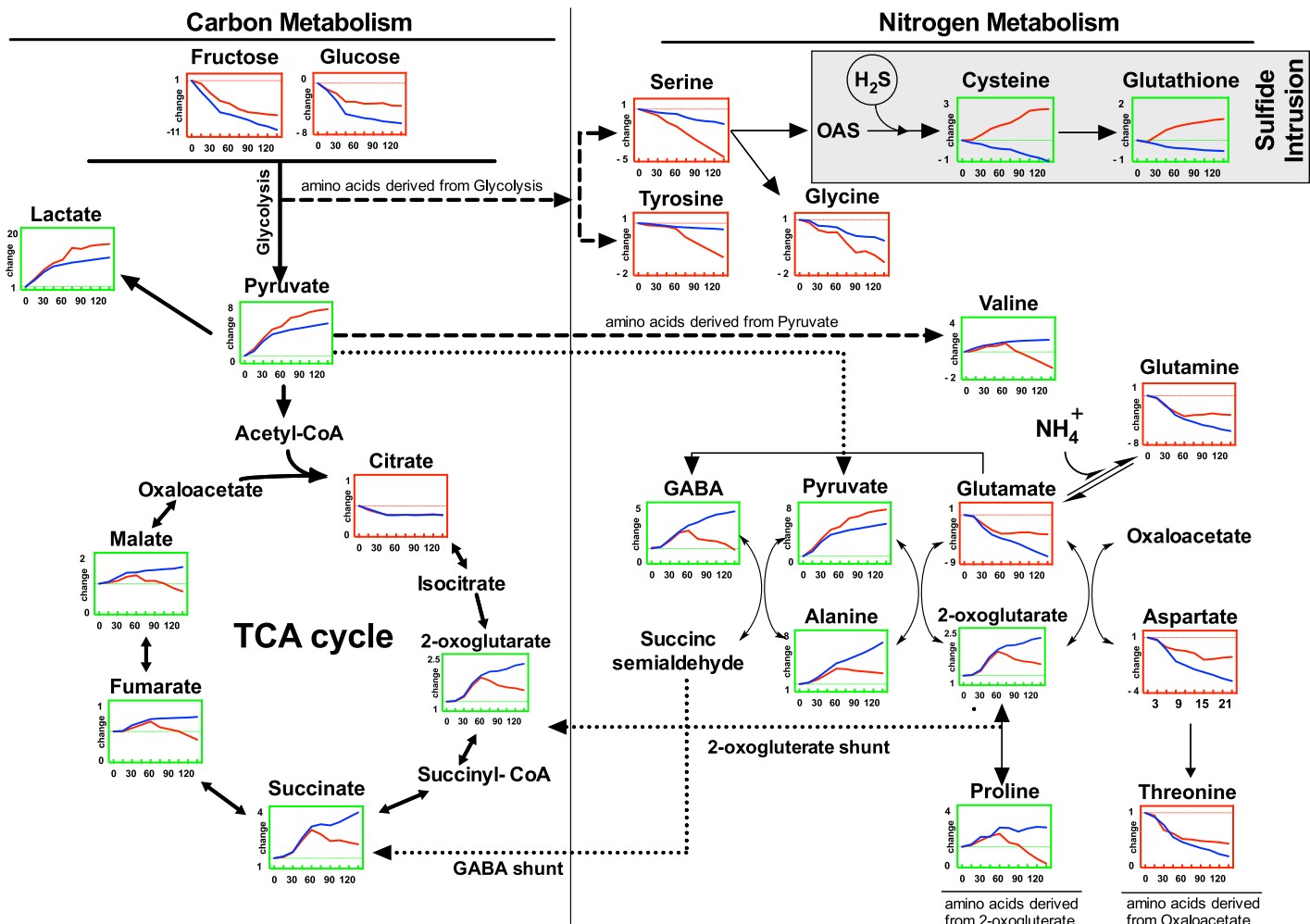

**Fig 3. Pathway analysis of the relative changes of metabolites in the leaves of *Z. marina* exposed to sulfide under dark hypoxic conditions.** Visualized are the glycolysis and TCA-cycle (left column), the associated nitrogen metabolism (right column) and sulfur intrusion pathway (grey box). The red line indicates data from sulfide exposed plants, and the blue line data from control plants. Metabolites levels are calculated relative to the data obtained from initial conditions before onset of darkness, hypoxia and sulfide exposure (T0), the time is denoted on the x-axis in minutes after onset of darkness, hypoxia and sulfide exposure. The metabolites framed in green showed significant increases upon hypoxia and the metabolites framed in red showed significant decreases (2-way ANOVA p < 0.05). Solid arrows illustrate enzymatic reactions, dotted lines indicate the same metabolite presented in different and linked pathways. Modified from Hasler-Sheetal, Castorani [22].

notable reduction in glucose and fructose levels (Fig 3). Furthermore, increased levels of pyruvate coupled with a decrease in citrate levels suggest a disruption in the normal metabolic pathway. Specifically, pyruvate appears to be restricted from entering the TCA cycle under hypoxia. This impediment leads to a limited carbon flux from pyruvate to citrate. Such a metabolic shift ultimately leads to energy deprivation stemming from the inhibition of oxidative energy metabolism in seagrasses under hypoxia [15]. Concurrently, there's a marked rise in in lactate levels pinpointing towards lactic fermentation. This process could potentially lead to cytosolic acidification a condition that can be detrimental to plant health, exhibiting phytotoxic effects [15, 16].

## Mitigation mechanisms under dark hypoxia in seagrasses

In response to the challenges of cytosolic acidification and energy deprivation during dark hypoxia, seagrasses have evolved a suite of mitigation strategies. Among these are: (1) The GABA Shunt: This pathway serves to counteract cytosolic acidification. (2) The 2-Oxoglutarate Shunt: This mechanism addresses the energy deficit by generating additional ATP and channeling surplus pyruvate into alanine. (3) Increased Glycolytic Flux (Pasteur Effect) [15]: This process enhances the breakdown of glucose to produce energy.

In line with these mitigation pathways, my observations revealed a decline in carbohydrate concentration. This was paired with an increase in lactate and pyruvate levels. Additionally, fluctuations in the levels of 2-oxoglutarate, gamma-aminobutyric acid (GABA), and alanine were observed (Fig 3). Such patterns suggest a depletion of carbohydrate pools due to increased glycolytic flux, a non-functioning TCA cycle resulting in energy depletion, and the activation of both the GABA and 2-oxoglutarate shunts. These shunts act as compensatory mechanisms to offset cytosolic acidification and energy deficiency during dark hypoxia.

Moreover, dark hypoxia triggered a reduction in amino acids derived from glycolysis and the TCA cycle, such as serine, tyrosine, glycine, and glutamine. In contrast, there was an accumulation of amino acids originating from pyruvate, including proline, valine, and alanine, within the *Z. marina* tissue (as shown in Fig 3). This perturbation of the amino acid profile underscores the suppression of oxidative energy metabolism and the activation of adaptive mechanisms to mitigate the effects of dark hypoxia.

## Sulfide exposure and poisoning

Sulfide exposure exhibited a significant impact on metabolic profile of seagrasses, particularly during latter stages of the experiment. Notably the initial phased (up to 45 minutes) appeared relatively unaffected by sulfide, but significant metabolic shifts were overserved post this period (Fig 3). This suggests that the primary impact of sulfide on carbon and nitrogen metabolism becomes evident after 45 minutes of exposure. When comparing the metabolic response under dark hypoxia alone to that of sulfide exposure a few key differences emerge: (1) Carbohydrate Metabolism: Sulfide exposure led to a less pronounced decline in carbohydrate levels, hinting at a reduced glycolytic flux. (2) Mitigation Mechanisms: Elevated levels of lactate and pyruvate were observed under sulfide exposure, indicating a potential dampening of the typical mitigation mechanisms seen under dark hypoxia. This was further supported by the observed decline in GABA, alanine, and 2-oxoglutarate levels post 45 minutes of sulfide exposure.

Seagrasses have evolved a dual strategy to counteract sulfide toxicity: tolerance and avoidance [1, 5, 14]. A key trait of this strategy under dark hypoxia is the detoxification of tissue-intruding phytotoxic sulfide by incorporating and converting it into non-toxic organic sulfur compounds, such as cysteine and glutathione [14, 26]. My observations corroborate this, as

sulfide exposure led to a swift and sustained increase of in cysteine and glutathione levels, alongside a decline in serine (Fig 3). This points to the rapid assimilation of sulfide into organic sulfur compounds as major sulfide tolerance pathway, a phenomenon previously noted in terrestrial plants [27] and hypothesized for seagrasses [14]. However the seagrass's ability to tolerate sulfide appears to decline after 45 minutes, suggesting that extended exposure cloud trigger tissue degradation.

Sulfide's primary mode of toxicity is its inhibitory effect on vital enzymes involved in both aerobic (cytochrome c oxidase) and anaerobic (alcohol dehydrogenase) energy metabolism [9], although other enzymes may also be affected [2]. Over time, the seagrass's capacity to tolerate sulfide seems to diminish, likely due to the depletion of essential substrates like serine and O-acetylserine. This is further compounded by the decline in glycolytic substrates, fructose and glucose, which are vital for replenishing serine and O-acetylserine pools (Fig 3) [28]. Additionally, the enzyme O-acetylserine(thiol)synthase (OASTL), crucial for converting sulfide into cysteine, may also be partially inhibited by sulfide poisoning (Fig 3). Given the high levels of sulfide (2 mM), it's plausible that phytotoxic sulfide persist in the tissue and the seagrass cannot detoxify it solely through incorporation [4, 5].

To investigate the effect of sulfide exposure on seagrass health, I analysed metabolites associated with senescence and tissue degradation. Phytol, a byproduct of chlorophyll breakdown and an indicator of plant senescence [29], showed a time-dependent increase post 45 minutes, suggesting detrimental effects of oxygen deprivation on leaf senescence (Fig 4). Remarkably, this senescence was further exacerbated during prolonged sulfide exposure, as indicated by a more pronounced increase in phytol levels in sulfide exposed leaves (Fig 4). Additionally, malondialdehyde (MDA), a marker for tissue degradation [30], increased in concentration in leaves after 60 minutes of sulfide exposure, a response not seen under dark hypoxia alone (Fig 4). This suggests that while dark hypoxia induces progressive leaf senescence, prolonged sulfide exposure (beyond 60 minutes) intensifies this process, leading to tissue degradation. The duration of tolerance during natural dark hypoxic events may vary based factors like individual sulfide resistance, overall health status, oxygen and light regimes and sulfide concentrations in seagrass ecosystems. [5, 14, 31].

## Conclusion

The study investigates the effects of sulfide intrusion on seagrasses, particularly under dark hypoxic conditions. Key findings and implications from the study are summarized as follows:

1. Rapid Sulfide Intrusion: The study confirms that sulfide rapidly permeates into seagrass leaves through the aerenchyma. This observation aligns with prior research that highlighted the swift depletion of oxygen in seagrasses under dark hypoxic conditions when studied in vitro [12].

2. Reduced Resilience to Dark Hypoxia: Sulfide exposure diminishes the ability of seagrasses to withstand dark hypoxic conditions. While seagrasses have evolved metabolic strategies to cope with sulfide intrusion, these mechanisms are effective only for a limited duration (approximately 45–60 minutes) under dark anoxic conditions. Beyond this timeframe, the protective pathways that mitigate the effects of hypoxia are inhibited.

3. Detrimental Effects on Seagrass Health: Prolonged exposure to sulfide not only disrupts the seagrass's metabolic responses to hypoxia but also triggers harmful outcomes. These include the onset of leaf senescence (aging) and tissue degradation, which can compromise the overall health and vitality of the seagrass.

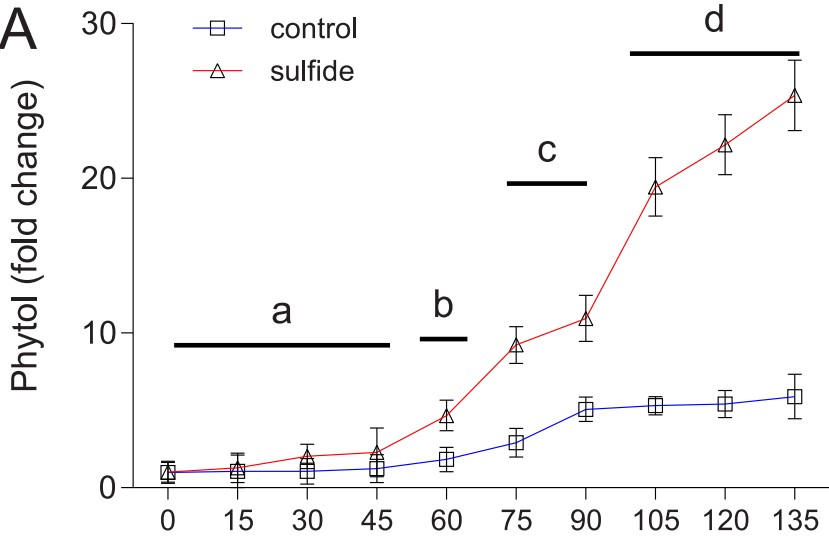

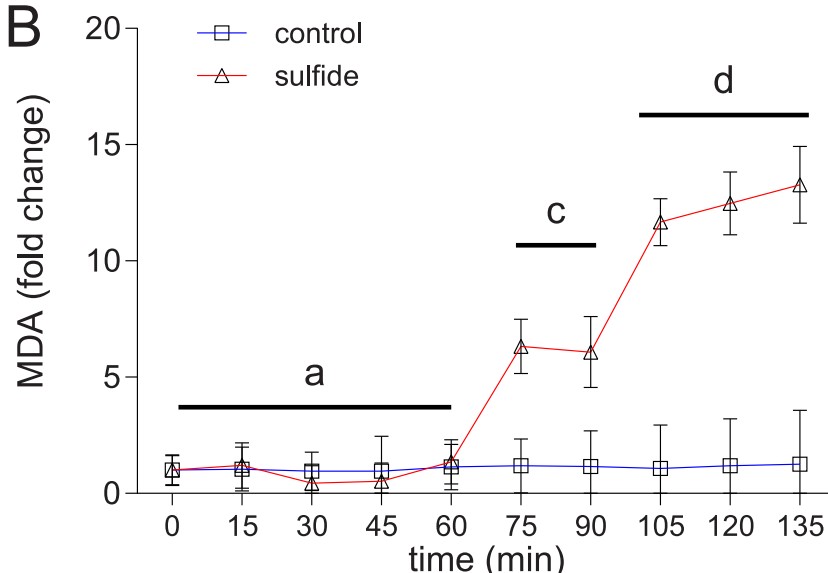

**Fig 4. Fold change of metabolites involved in tissue degradation.** (A) The upper panel shows phytol–an indicator for leave senescence; and (B) the lower panel malondialdehyde (MDA)–an indicator for tissue degradation–in *Zostera marina* leaves exposed to sulfide intrusion (triangles) and controls (squares). Metabolites levels are calculated as fold-change relative to the data obtained from initial conditions before onset of darkness, hypoxia and sulfide exposure (T0), the time is denoted on the x-axis in minutes after onset of darkness, hypoxia and sulfide exposure. Values represent mean ± SEM. Levels not sharing the same letter indicate significant differences (ANOVA; $p < 0.05$; Tukey's posthoc test, $p < 0.05$).

4. Future Implications for Seagrass Ecosystems: The global trend of increasing eutrophication in estuaries and coastal regions poses a significant threat to seagrass ecosystems. Eutrophication leads to enhanced hypoxic conditions and diminished light availability for seagrasses [32, 33]. As a result, seagrass habitats are likely to experience more frequent and intense

episodes of sulfide poisoning. This can have cascading effects on the broader marine ecosystem, given the pivotal role seagrasses play in coastal environments.

In conclusion, the study underscores the vulnerability of seagrasses to sulfide intrusion, especially in the backdrop of changing environmental conditions. It emphasizes the need for conservation efforts to mitigate the impacts of eutrophication and safeguard the health and sustainability of seagrass ecosystems in estuaries worldwide.

## Supporting information

**S1 Data.**
(XLSX)

**S1 File.**
(DOCX)

**S2 File.**
(PDF)

## Acknowledgments

I would like to express my gratitude to Marianne Holmer, Erik Laursen, and Mia Østergaard Pedersen for their valuable research assistance and insightful discussions. I also extend my thanks to the University of Southern Denmark (SDU) for providing the facilities for this study.

## Author Contributions

**Conceptualization:** Harald Hasler-Sheetal.

**Writing – review & editing:** Harald Hasler-Sheetal.

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
