## [Decision Letter · Decision Letter 0]

10 Sep 2023

PONE-D-23-22011Detrimental impact of sulfide on the seagrass Zostera marina in dark hypoxiaPLOS ONE

Dear Dr. Hassler-Sheetal,

Thank you for submitting your manuscript to PLOS ONE. After careful consideration, we feel that it has merit but does not fully meet PLOS ONE’s publication criteria as it currently stands. Therefore, we invite you to submit a revised version of the manuscript that addresses the points raised during the review process.

We look forward to receiving your revised manuscript.

Kind regards,

Vasu D. Appanna

Academic Editor

PLOS ONE

A clean copy of the edited manuscript (uploaded as the new *manuscript* file).

“I would like to express my gratitude to Marianne Holmer, Erik Laursen, and Mia Østergaard Pedersen for their valuable research assistance and insightful discussions. I also extend my thanks to the University of Southern Denmark (SDU) for providing the funding for this study. I declare that none of the authors have any conflicts of interest.”

“The author received no specific funding for this work.”

6. Please ensure that you refer to Figures 3 and 4 your text as, if accepted, production will need this reference to link the reader to the figure.

Additional Editor Comments:

Dear Dr. Hasler-Sheetal

Although this study is of interest, there are numerous revisions that have to be undertaken. Both reviewers have identified shortcomings that need to be addressed. It is essential to indicate the statistical significance of the experiments performed so that the results can be validated.

Best wishes

Vasu

Vasu D. Appanna PhD

Reviewers' comments:

Reviewer's Responses to Questions

**Comments to the Author**

1. Is the manuscript technically sound, and do the data support the conclusions?

Reviewer #1: Yes

Reviewer #2: No

2. Has the statistical analysis been performed appropriately and rigorously? 

Reviewer #1: No

Reviewer #2: No

3. Have the authors made all data underlying the findings in their manuscript fully available?

Reviewer #1: Yes

Reviewer #2: No

4. Is the manuscript presented in an intelligible fashion and written in standard English?

Reviewer #1: No

Reviewer #2: Yes

5. Review Comments to the Author

Reviewer #1: This is an interesting manuscript describing the impact of hydrogen sulfide on Zostera marina in the dark and hypoxic conditions. This study was carried out in a seawater aquarium. Therefore, it is important to specify in the abstract, discussion and conclusion sections that the study was carried out in vitro. The statistical significance values (e.g. p values) are required to compare the control and experimental dataset. The English language of the manuscript requires editing.

The following points need to be addressed to improve the quality of the manuscript.

Specific comments:

* Introduction: Please specify the rational for selection of Zostera marina along with the morphotype used in this study.

* Lines 100: The state (solid or liquid) of Na2S that was added should be specified here.

* Line: 101: Sulfide-free..? Was the chamber 100% free from sulfide? How was the sulfide concentration measured?

* Line 150-151: “…low micromolar levels”: Give the numerical values here.

* Line 153-154: Specify the normal physiological sulfide level here.

* Line 170-172: Give statistical significance value (e.g. p value) to compare the control and experimental dataset.

* Line 172-173: Provide statistical evidence for significant impact (e.g. p values).

* Line 269-273 & Line 273-277: Give statistical significance values (e.g. p value) to compare the control and experimental dataset.

* Conclusion: It is essential to specify here that the experiments were conducted in vitro.

Reviewer #2: This is an interesting and well-written study on the harms of eutrophication in seagrasses. Unfortunately, it is difficult to know if the experiments were conducted with rigor as there is a lack of statistical analysis and the metabolomics dataset does not appear to be available for review.

- There are no figure captions for figure 3 or 4.

- Figure 3 y axis is listed as either change or ratio. Why is that?

- No error bars for any of the metabolites or statistical analysis, rendering it hard to determine if any of the data is meaningful.

- Are the changes in phytol or MDA significant? What statistical analyses have you done? How were these experiments performed, as I do not see this in the methodology?

6. PLOS authors have the option to publish the peer review history of their article (what does this mean?). If published, this will include your full peer review and any attached files.

Reviewer #1: No

Reviewer #2: No

---

## [Author Response · Author response to Decision Letter 0]

1 Nov 2023

Dear Academic Editor Dr. Vasu D. Appanna and Reviewers,

First and foremost, I sincerely apologize for the incomplete data in the submitted manuscript. This oversight occurred due to an error during the final preparation of the documents. Such an oversight should not have happened, and I regret that it was not caught during the proofreading phase. I deeply appreciate your understanding and patience.

Thank you for your constructive feedback on my manuscript titled "Detrimental impact of sulfide on the seagrass Zostera marina in dark hypoxia." I appreciate the time and effort you have dedicated to reviewing my work. I have carefully addressed each comment and made the necessary revisions to improve the clarity and quality of my manuscript. Below, I first provide a general response followed by a point-by-point response to the comments and concerns raised.

Reviewer #1 Comments and Responses:

1. Comment (Abstract, Discussion, Conclusion sections): The study was carried out in a seawater aquarium. Therefore, it is important to specify in the abstract, discussion, and conclusion sections that the study was carried out in vitro. 

Response: I acknowledge this oversight and have now explicitly mentioned in the abstract, discussion, and conclusion sections that the study was conducted in vitro.

2. Comment (Introduction): Specify the rationale for the selection of Zostera marina along with the morphotype used in this study.

Response: I appreciate the suggestion. The rationale for selecting Zostera marina L. is its ecological significance and widespread presence in the temperate Northern Hemisphere, especially in Denmark. I had elaborated on this in the introduction, emphasizing its relevance for my study. The specific morphotype from the Danish straits was chosen due to its exposure to high sulfide levels, a consequence of eutrophication in the region.

3. Comment (Line 100): Specify the state (solid or liquid) of Na2S that was added.

Response: I apologize for the oversight. The sulfide solutions were prepared from Na2S nonahydrate in its solid form. I have now clarified this in the methods section.

4. Comment (Line 101): Was the chamber 100% free from sulfide? How was the sulfide concentration measured? 

Response: In my experiments, the "sulfide-free" chambers had sulfide levels below the detection limits of the Cline method, which I used for measurement. I have had provided this clarification in the manuscript.

5. Comment (Lines 150-151): Provide numerical values for "low micromolar levels." 

Response: I have amended the manuscript to specify the range as 10-100µM.

6. Comment (Lines 153-154): Specify the normal physiological sulfide level. 

Response: I have added the observed levels, which were around 325µM.

7. Comment (Lines 170-172, 172-173, 269-273 & 273-277): Provide statistical significance values (e.g., p-values) for various datasets. 

Response: I have now included the missing statistical information, including p-values, in the manuscript. Additionally, the supplementary material now contains the raw data and ANOVA test-statistics. In the context of PCA (Fig2), p-values are not typically calculated because the analysis does not involve a probabilistic model that allows for the computation of statistical significance in the way that, say, a t-test or ANOVA would. The focus is rather on the explained variance by each principal component, which is a different concept from statistical significance.

8. Comment (Conclusion): Specify in the conclusion that the experiments were conducted in vitro. Response: I concur with your suggestion and have made the necessary amendments in the conclusion section.

Reviewer #2 Comments and Responses:

1. Comment: There are no figure captions for figure 3 or 4. 

Response: I apologize for this oversight. The missing captions have been added to the revised manuscript.

2. Comment (Figure 3): Y-axis is listed as either change or ratio. Why is that? 

Response: The values represent fold-change. I have now clarified this in the figure caption and corrected the clerical error.

3. Comment (Figure 3): No error bars for any of the metabolites or statistical analysis. 

Response: For clarity in the pathway analysis (Figure 3), I omitted error bars. However, based on your feedback, I have now provided the underlying data, 2-way ANOVA test-statistics, and illustrations of all metabolites with their replicates in the supplementary material.

4. Comment: Are the changes in phytol or MDA significant? What statistical analyses have you done? How were these experiments performed, as I do not see this in the methodology? 

Response: The changes in phytol and MDA were indeed significant. I have now presented the post-hoc test results in the figure and its caption. The raw data and ANOVA test-statistics have also been added to the supplementary material for further clarity.

I believe that the revisions made address the concerns raised by the reviewers and enhance the quality of my manuscript substantially.

I hope that my responses and the changes made to the manuscript will make it suitable for publication in PLOS ONE.

Thank you for considering my work.

Sincerely,

Dr. Hasler-Sheetal

---

## [Decision Letter · Decision Letter 1]

22 Nov 2023

Detrimental impact of sulfide on the seagrass Zostera marina in dark hypoxia

PONE-D-23-22011R1

Dear Dr. Hasler-Sheetal,

We’re pleased to inform you that your manuscript has been judged scientifically suitable for publication and will be formally accepted for publication once it meets all outstanding technical requirements.

Kind regards,

Vasu D. Appanna

Academic Editor

PLOS ONE

Additional Editor Comments (optional):

Reviewers' comments:

Reviewer's Responses to Questions

**Comments to the Author**

1. If the authors have adequately addressed your comments raised in a previous round of review and you feel that this manuscript is now acceptable for publication, you may indicate that here to bypass the “Comments to the Author” section, enter your conflict of interest statement in the “Confidential to Editor” section, and submit your "Accept" recommendation.

Reviewer #1: All comments have been addressed

Reviewer #2: All comments have been addressed

2. Is the manuscript technically sound, and do the data support the conclusions?

Reviewer #1: (No Response)

Reviewer #2: Yes

3. Has the statistical analysis been performed appropriately and rigorously? 

Reviewer #1: (No Response)

Reviewer #2: No

4. Have the authors made all data underlying the findings in their manuscript fully available?

Reviewer #1: (No Response)

Reviewer #2: Yes

5. Is the manuscript presented in an intelligible fashion and written in standard English?

Reviewer #1: (No Response)

Reviewer #2: Yes

6. Review Comments to the Author

Reviewer #1: (No Response)

Reviewer #2: Figure 4 should demonstrate whether there is a significant difference between control and sulfide groups, not within the same group. Otherwise, I accept the author's revisions otherwise.

7. PLOS authors have the option to publish the peer review history of their article (what does this mean?). If published, this will include your full peer review and any attached files.

Reviewer #1: No

Reviewer #2: No

---

## [Editor Report · Acceptance letter]

28 Nov 2023

PONE-D-23-22011R1 

Detrimental impact of sulfide on the seagrass *Zostera marina* in dark hypoxia 

Dear Dr. Hasler-Sheetal:

I'm pleased to inform you that your manuscript has been deemed suitable for publication in PLOS ONE. Congratulations! Your manuscript is now with our production department. 

Kind regards, 

on behalf of

Dr. Vasu D. Appanna 

Academic Editor

PLOS ONE